# Perturbations of Zinc Homeostasis and Onset of Neuropsychiatric Disorders

**DOI:** 10.3390/ijms262210877

**Published:** 2025-11-09

**Authors:** Gavino Faa, Carlotta Meloni, Mara Lastretti, Martina Pinna, Mirko Manchia, Pasquale Paribello

**Affiliations:** 1Department of Medical Sciences and Public Health, University of Cagliari, 09042 Cagliari, Italy; gavinofaa@gmail.com; 2Department of Biology, College of Science and Technology, Temple University, Philadelphia, PA 19122, USA; 3Section of Psychiatry, Department of Medical Sciences and Public Health, University of Cagliari, 09124 Cagliari, Italy; c.meloni45@studenti.unica.it (C.M.); pasquale.paribello@unica.it (P.P.); 4Department of Economic, Psychological and Communication Sciences, Niccolò Cusano University, 00166 Rome, Italy; mara.lastretti@unicusano.it; 5Forensic Psychiatry Unit, Department of Mental Health and Addiction, Health Agency of Cagliari, 09124 Cagliari, Italy; martina.pinna@aslcagliari.it; 6Department of Pharmacology, Dalhousie University, Halifax, NS B3K 6R8, Canada

**Keywords:** metallomics, homeostasis, psychiatric disorders, neuroprotection, neurodevelopment, risk prediction

## Abstract

Zinc (Zn^2+^) is a trace element essential for its catalytic, antioxidant, and immunomodulatory roles extending to synaptic signalling in the central nervous system. In this narrative review, we aim to offer the reader evidence linking perturbations of the Zn^2+^ homeostasis, including deficiency, excess, or transportation anomalies, to neuropsychiatric conditions such as Alzheimer’s disease (AD), Parkinson’s disease (PD), autism spectrum disorder (ASD), attention deficit hyperactivity disorder (ADHD), schizophrenia (SCZ), major depressive disorder (MDD), and bipolar disorder (BD). A targeted, unsystematic PubMed search followed by an extensive pearl-growing strategy was applied to further augment study selection based on the extensive expertise of study authors. Overall, most of the evidence currently available suggests a modest benefit for a Zn^2+^ supplement of around 25–30 mg/day as an augmentation to MDD treatment, with potential benefits of smaller magnitude in paediatric ADHD. Evidence for perturbations of Zn^2+^ as a biomarker of risk for these neuropsychiatric disorders remains unconvincing. The role of Zn^2+^ supplements in the treatment of the selected conditions remains largely unknown due to the lack of specific, randomised controlled trials conducted to explore their efficacy. The long-term safety, optimal doses for specific applications, and the exploration of possible biomarkers to stratify patient selection to identify the optimal candidate for Zn^2+^ supplements remain unanswered questions.

## 1. Introduction

Zinc (Zn^2+^) is an essential trace element fundamental for human life, participating in all aspects of metabolism, growth, development, immune functions, and neurophysiology [1]. Zn^2+^ has an atomic weight of 65.39, a solubility of 4 mg/L, and has no redox activity. In human physiology, Zn^2+^ plays a key structural role in multiple enzymes, in which Zn^2+^ ions are incorporated tetrahedrally into characteristic amino acids, giving rise to coordinative bonds for the stability of multiple proteins, such as disulfide isomerase, involved in vital processes [2]. Zn^2+^ is a cofactor of numerous transcription factors and over 300 metalloenzymes, playing a crucial role in regulating multiple physiological processes, including anti-inflammatory, antioxidant, and immune responses [3,4]. In a group of enzymes, including carboxypeptidase, Zn^2+^ ions participate directly in enzyme catalysis and, in their absence, the enzymes become inactive [5]. Zn^2+^ is a cofactor of the cysteine-rich intestinal protein (CRIP), which regulates T-helper function and the secretion of multiple interleukins and interferon-gamma. Zn^2+^ deficiency influences CRIP function, causing a dysregulation of cytokines required for host defence [6]. Zn^2+^ is involved in DNA replication and transcription; moreover, it is estimated that about 1% of the human genome encodes for Zn^2+^ finger proteins [7].

In the central nervous system (CNS), Zn^2+^ ions act as structural and regulatory catalysts for all major classes of enzymes, stabilise Zn^2+^-finger proteins involved in transcriptional regulation, and function as an intracellular signalling molecule [8]. Zn^2+^ plays an additional role in the CNS, acting as a neurosecretory cofactor in a subset of Zn^2+^-containing glutamatergic neurons [9]. Free ionic Zn^2+^ is present in neurons of the cortex, hippocampus, amygdala, and olfactory bulbs [10]. The role of Zn^2+^ in the human brain has been debated in recent years. Being a trace element, discrete quantities of Zn^2+^ are fundamental in brain physiology, whereas Zn^2+^ overload may cause neurotoxic damage to postsynaptic neurons [11]. On the other hand, Zn^2+^ deficiency has been reported as a causative factor in the decrease in pituitary function in experimental animals [12]. Given the pleiotropic effects of Zn^2+^ on the CNS and the possible role of perturbations of its homeostasis in neuropsychiatric disorders, this review is poised to summarise the evidence linking Zn^2+^ to the risk of neuropsychiatric disorders, focusing on the role of Zn^2+^-containing neurons in the human cerebral cortex, both in physiological and pathological processes [10].

## 2. Methods

In the following sections we provide a selection of the most relevant papers exploring the association of plasma or serum Zn^2+^ levels in Parkinson’s disease (PD), Alzheimer’s disease (AD), major depressive disorder (MDD), attention deficit hyperactivity disorder (ADHD), autism spectrum disorder (ASD), schizophrenia (SCZ), and bipolar disorder (BD). We augmented the scope of the review itself by carrying out a targeted PubMed search with the following search strategy: “(“Zn^2+^”[Mesh] OR Zn^2+^[tiab] OR “serum Zn^2+^”[tiab] OR “plasma Zn^2+^”[tiab] OR “Zn^2+^ intake”[tiab] OR “dietary Zn^2+^”[tiab] OR “Zn^2+^ supplementation”[tiab] OR “Zn^2+^ sulfate”[tiab]) AND (“Depression”[Mesh] OR depression[tiab] OR “Major Depressive Disorder”[tiab] OR “Bipolar Disorder”[Mesh] OR bipolar[tiab] OR “Schizophrenia”[Mesh] OR schizophrenia[tiab] OR “Autism Spectrum Disorder”[Mesh] OR autism[tiab] OR ASD[tiab] OR “Attention Deficit Disorder with Hyperactivity”[Mesh] OR ADHD[tiab] OR “attention deficit hyperactivity”[tiab] OR “Parkinson Disease”[Mesh] OR parkinson*[tiab] OR “Alzheimer Disease”[Mesh] OR alzheimer*[tiab]) AND (randomized controlled trial[pt] OR clinical trial[pt] OR random*[tiab] OR placebo[tiab] OR cohort[tiab] OR “case-control”[tiab] OR “cross-sectional”[tiab] OR observational[tiab] OR epidemiolog*[tiab]) AND (“Humans”[Mesh])”. Furthermore, an extensive pearl-growing strategy was carried out by analysing the references for review papers and all relevant research articles exploring the researched topic. Two authors (P.P. and M.M.) were responsible for the narrative extraction of results from the identified studies.

## 3. The Role and Zn^2+^ Levels in Neurodevelopmental Disorders

### 3.1. The Role and Level of Zn^2+^ in ADHD

Several studies have tested whether changes in Zn^2+^ levels were associated with neurodevelopmental disorders. One study on trace element concentration carried out in children with ADHD measured the content of essential trace elements in hair samples, identifying a statistically significant deficiency of Zn^2+^ associated with illness status [13]. A meta-analysis on Zn^2+^ status in ADHD carriers reported conflicting results, particularly on the value of Zn^2+^ content in hair samples. On the other hand, the meta-analysis raised the possibility that children with ADHD might be prone to Zn^2+^ deficiency, indicating that screening for Zn^2+^ levels could be justified in these children [14]. A more recent study focused on a relatively large cohort (895 subjects) of children and adolescents with behavioural problems, showing an inverse linear relationship between serum Zn^2+^ and Mg^2+^ levels and the prevalence of behavioural problems [15]. These findings corroborated the hypothesis that Zn^2+^ status might play a relevant role in modulating the risk and progression of behavioural problems and emphasise the importance of screening programmes devoted to maintaining adequate levels of Zn^2+^, particularly in childhood [16]. Table 1 summarises the results of the observational studies investigating the potential association of Zn^2+^ levels and ADHD or its association with specific ADHD symptoms.

Several observational studies to date have explored the association between Zn^2+^ levels and ADHD. A case–control study [17] compared children with ADHD with age-matched controls, finding lower Zn^2+^ concentrations in the former but no association with methylphenidate treatment. An additional report corroborated these earlier findings, showing that children with ADHD might have lower serum Zn^2+^ concentrations compared to healthy controls [18]. In this study, Zn^2+^ deficiency was also correlated with a higher symptom severity [18], whilst an additional paper reported higher hair levels of Zn^2+^ to be associated with a higher burden of inattentive, hyperactive, and total ADHD symptom dimensions [19]. Additional studies explored Zn^2+^ levels in association with ADHD treatment and possible comorbidities. A randomised controlled trial of methylphenidate withdrawal [20] assessed the association of Zn^2+^ levels with clinical outcomes. Higher baseline Zn^2+^ levels were associated with poorer performance on a working memory task following discontinuation. A study on type 1 diabetes mellitus in children with ADHD compared to children with only type 1 diabetes found that the glycosylated haemoglobin and copper/Zn^2+^ ratio correlated positively with symptom severity, but only in children with ADHD comorbidity, suggesting a possible role of Zn^2+^ dysregulation in this subpopulation [21]. Zn^2+^ dietary deficit was also described in children with ADHD and type 1 diabetes mellitus as compared with children with only type 1 diabetes, adding a potential mechanism for Zn^2+^ anomalies in this population [22]. Robinson et al. [23], in 2024, described an increased risk for ADHD combined type in association with salivary Zn^2+^ levels [24]. Interventional studies have also been conducted to explore Zn^2+^ supplementation in the treatment of ADHD. Table 2 summarises findings of interventional studies of Zn^2+^ supplements in ADHD. Indeed, Bilici et al. [24] reported that a 12-week Zn^2+^ sulphate monotherapy resulted in significantly reduced hyperactivity and impulsivity scores, along with an expected rise in Zn^2+^ blood levels [25]. Adjunctive Zn^2+^ sulphate to methylphenidate also improved ADHD ratings from both parents and teachers according to an additional report [25]. An additional trial from the United States [26] tested Zn^2+^ glycinate augmentation to amphetamine treatment, finding that supplementation permitted a reduction in amphetamine dose. An additional report further corroborates the Zn^2+^ adjunct as an effective addition to standard pharmacotherapy [27].

### 3.2. The Role and Level of Zn^2+^ in ASD

Additional papers on neurodevelopmental conditions have explored the potential association of Zn^2+^ in ASD. One study on trace element concentration carried out in children with ASD, measuring the content of essential trace elements in hair samples, found that the most deficient element was Zn^2+^ [28]. Zn^2+^ deficiency was found in 92% of affected children as compared with 20% of healthy subjects [28]. A case–control study assessed trace elements in hair and nail samples, suggesting significant variability in Zn^2+^ levels, especially among individuals with lower severity autism, suggesting Zn^2+^ imbalance may relate to symptom expression within the spectrum [29]. An additional paper showed that ASD patients had lower Zn^2+^ levels and lower Zn^2+^/copper ratios compared to healthy controls. Lower Zn^2+^/copper ratios were also associated with greater symptom severity [30]. A further case–control study from Macedoni-Lukšič reported a significantly elevated blood copper/Zn^2+^ ratio relative to control neurological conditions but with no different urinary porphyrin profile [31]. Table 3 summarises findings of studies of Zn^2+^ levels in ASD.

## 4. Zn^2+^ Levels and Parkinson’s Disease

Changes in Zn^2+^ homeostasis have been implicated in PD. Overactivation of the cortico-striatal glutamatergic system underlies the development of dopaminergic toxicity. Since Zn^2+^ ions act as a synaptic transmitter in the brain, alterations of vesicular or synaptic Zn^2+^ signalling in the basal ganglia could contribute to the onset and progression of PD [32]. The functional relationship between Zn^2+^ levels and the glutamatergic system could be explained by the link between Zn^2+^ and NMDA receptors, the best characterised synaptic targets of Zn^2+^ ions [33]. Recently, the relationship between Zn^2+^ ions and NMDA-type glutamate receptors has been implicated in ASD [34]. Table 4 summarises the results from the single observational study exploring Zn^2+^ levels in PD. Evidence linking Zn^2+^ to PD appears entirely based on observational studies. Forsleff et al. [35], back in 1999, used a Zn^2+^ taste test and serum indices in approximately 100 individuals living with PD and 25 controls, with PD patients showing a significantly decreased Zn^2+^ status compared to controls, along with health-related variables thought to be related to Zn^2+^ status (vision problems, olfactory loss, and taste loss) [35]. A meta-analysis from Sun et al. [36] reported that lower Zn^2+^ levels were significantly associated with a higher risk of PD [36]. An additional review from Zhao et al. [37] further supported this line of evidence, describing lower levels of serum Zn^2+^ and higher levels of hair Zn^2+^ as associated with a higher risk of PD [37]. Finally, a large case–control study from China, including 238 patients living with PD and 302 age- and sex-matched controls [38], found reduced Zn^2+^ concentrations in patients compared with controls. We found no specific interventional study addressing Zn^2+^ as a supplement in the management of PD.

## 5. Alzheimer’s Disease

AD and PD are the most frequent age-related neurodegenerative disorders, with perturbations in Zn^2+^ and copper homeostasis playing a pivotal role in their pathogenesis [39,40,41,42]. Table 5 summarises the main findings from studies of Zn^2+^ levels in AD. Three studies focused on the potential role of Zn^2+^ in AD. A postmortem study [43] showed that brain samples had a reduced expression of Zn^2+^ transporter 3 (ZnT3) in the dorsolateral prefrontal cortex, and this was also associated with depression severity across three dementia groups, including AD, suggesting a role for synaptic Zn^2+^ perturbations in neuropsychiatric symptoms of the disease. Additional reports in live patients have shown more nuanced results in this area. Rembach et al. [44] reported that serum Zn^2+^ concentrations dropped with advancing age and that the previously observed association between Zn^2+^ and AD could instead be better explained by its association with age [44]. In line with these findings, an additional report [45] described no significant differences in Zn^2+^ levels between individuals living with AD and controls, but observed a sex-specific trend for higher levels of Zn^2+^ in male patients.

## 6. Schizophrenia

We included one paper investigating the association of Zn^2+^ derangements with SCZ. This case–control study from China [46] matched 114 patients with 114 sex- and age-matched controls, with no significant results described for the comparison of Zn^2+^ levels between patients and control. In Table 6, the results of this study are illustrated.

## 7. Bipolar Disorder

Three studies explored potential differences in Zn^2+^ metabolism between patients with BD and controls (Table 7), with contrasting results. Specifically, a cross-sectional study from Poland [47] found that individuals living with BD type I during a depressive episode featured lower serum Zn^2+^ levels than during other mood states, suggesting that Zn^2+^ levels may fluctuate with mood states. An additional study from Sweden [48] comprising individuals living with BD described no significant association between Zn^2+^ serum levels and inflammatory status, executive functions, or severity of symptoms. An additional study from Chebieb et al. [49] described lower plasma Zn^2+^ levels in individuals with BD compared to healthy controls, along with a correlation between lipid peroxidation and the Cu^2+^/Zn^2+^ ratio, suggesting a possible role for trace element imbalance in influencing oxidative stress [49].

## 8. Major Depressive Disorder

Arguably, the most convincing evidence in the field of mental health for the role of Zn^2+^ in neuropsychiatric conditions is in MDD. We included eight original research articles describing the results of randomised clinical trials (Table 8) and five observational studies (Table 9). Early studies from Poland [50,51] presented contradictory results, with the first and smaller from the group of Nowak suggesting a potential benefit of Zn^2+^ supplement in reducing depressive symptoms, and with the following and larger study from Siwek et al. finding no benefit, with the possible exception of individuals with treatment-resistant depression [50,51]. An additional study from Japan [52] found that the addition of Zn^2+^ supplement in women taking multivitamins might improve mood-related subscales which were, unsurprisingly, associated with increased Zn^2+^ levels. Five additional studies, all conducted in Iran [53,54,55,56,57], all found significantly greater mood improvement with Zn^2+^ addition. Zn^2+^ augmentation to selective serotonin reuptake inhibitors led to improvements in mood symptoms compared to placebo in two trials from the group of Ranjbar [53,54]. No changes in BDNF or inflammatory markers were observed in these reports. Salari et al. [55] described Zn^2+^ as a possible viable augmentation for individuals with multiple sclerosis and comorbid MDD, but no effect on neurological symptoms was observed. An additional randomised controlled [56] trial focused on individuals overweight or obese with MDD and found that Zn^2+^ monotherapy might be effective in reducing depressive symptoms and promoting an increase in BDNF. An additional factorial trial [57] combining Zn^2+^ and vitamin D supplementation found that Zn^2+^ supplementation was associated with greater mood improvements but with no effect on BDNF or cortisol.

Observational studies consistently reported a relatively lower level of Zn^2+^ status compared to controls. Maes et al. [58] reported that among individuals with MDD, serum Zn^2+^ and albumin appear significantly lower as compared to healthy controls, suggesting that hypoalbuminemia may at least in part play a role in explaining the observed peripheral reduction in Zn^2+^ concentrations. Data from the Boston Area Community Health survey [59] provides further insight into this association, reporting that low dietary Zn^2+^ was associated with a greater burden of depressive symptoms among women but not men. In contrast, a large cohort study from Finland [60] with a two-decade follow-up probing the association between dietary Zn^2+^ intake and the subsequent risk of hospital discharge diagnosis of MDD was negative after excluding individuals with elevated depressive symptoms at baseline. Furthermore, a case–control study from Poland [61] found that those achieving symptomatic remission showed Zn^2+^ concentrations similar to healthy controls, suggesting a potential link between Zn^2+^ status and treatment response. Finally, an additional case–control study from Islam et al. [62] found among individuals with MDD significantly lower Zn^2+^, along with lower magnesium, calcium, manganese, selenium, and increased copper.

## 9. Discussion

In this narrative review, we have reviewed how alterations of Zn^2+^ homeostasis—either excess, deficiency, or anomalies in its metabolism—might modulate the risk of developing neuropsychiatric disorders. The involvement of Zn^2+^ in over 300 metalloenzymes as well as in multiple physiological pathways [3,4] may provide a plausible explanation for the link between peripheral Zn^2+^ levels, central neurotransmission, oxidative stress, and neuroinflammation. Overall, the most convincing evidence for the role of Zn^2+^ supplements is available for the management of MDD. Indeed, a meta-analysis showed clearly that the concentration of Zn^2+^ in the peripheral blood of depressed patients is approximately 1.850 mmol/L lower than that of control subjects [63]. Interestingly, while most of the included studies reported the means of depressed and control groups to be within normal laboratory reference ranges, patients with MDD means were often near the lower boundary of the normal range [63]. Particularly intriguing might be the observation of a more pronounced effect in severe treatment-resistant depression. A parallel insight comes from a recent paper [64] in treatment-resistant depression: rather than focusing solely on the global state of treatment resistance, the study explicitly recruited patients with a defined mutation in the *ANK3* gene, finding that in this subpopulation, the study medication was more effective than placebo in reducing depressive symptoms, despite the failure of the liafensine itself in the non-selected treatment-resistant depression population. Preliminary evidence indicates that Zn^2+^ represents a valuable augmentation option to existing treatment for MDD, with at least one trial suggesting a potential benefit for treatment-resistant depression [51]. Exploring possible biological determinants for Zn^2+^ supplement efficacy for this indication for a subpopulation of treatment-resistant depression, rather than tapping on treatment-resistant as an individual, global state, may represent a particularly valuable prospect, especially considering the overall safety and how inexpensive Zn^2+^ supplements are (Table 10 summarises recommended levels for Zn^2+^ and possible side effects).

The doses typically reported in augmentation studies appear to be well below levels which could be potentially dangerous or indeed difficult to tolerate. The “Clinician guidelines for the treatment of psychiatric disorders with nutraceuticals and phytoceuticals” from the World Federation of Societies of Biological Psychiatry (WFSBP)/Canadian Network for Mood and Anxiety Treatments (CANMAT) Taskforce (2022) endorse with a “provisional recommended” indication for the introduction of Zn^2+^ supplement in MDD [66]. There is, however, insufficient evidence of Zn^2+^ supplement in the management of ADHD [66]. At present, the evidence for the potential role of Zn^2+^ as either a biomarker or as a potential treatment for the remaining conditions analysed in the present review appears too limited to make any definitive assessment. Alterations of Zn^2+^ homeostasis are biologically plausible and appear relevant across several neuropsychiatric conditions, but the most convincing clinical signal to date concerns MDD. Zn^2+^ supplementation shows promise as adjunctive therapy in MDD, with a smaller but possible efficacy signal in paediatric ADHD. However, evidence remains inconclusive for the other neuropsychiatric disorders (PD, AD, BD, ASD, and SCZ). Interestingly, novel approaches could include the production of novel zinc-based drug substances that may show more effective properties in modulating the biological perturbations associated with neuropsychiatric disorders. Indeed, a recent study used KLS-1, which is zinc aspartate enriched with light isotope ^64^Zn to 99.2% atomic fraction of total zinc, in an animal model of AD, showing a decreased inflammatory load in the CNS of rats that correlated with the improvement of short-term spatial memory and cognitive flexibility, and moderately with the betterment of remote spatial memory [67].

Finally, we should highlight that this review is limited by the application of a non-systematic approach. Thus, evidence pertinent to the role of Zn^2+^ in neuropsychiatric disorders might not have been identified exhaustively, making our indications preliminary and in need of integration with other sources of data. To move the field forward, biomarker-stratified randomised trials—standardising biospecimens and incorporating measures such as baseline Zn^2+^, Cu/Zn ratio, albumin/CRP, and transporter genetics—are warranted to identify which patients are most likely to benefit, while monitoring safety (notably copper status) during supplementation.

## Figures and Tables

**Table 1 ijms-26-10877-t001:** Summary of observational studies exploring Zn^2+^ levels in ADHD and their association with symptom severity.

Author, Year	Country	Sample Features (Mean Age—Range, % Female)	Design	Reported Outcomes	Results
Toren et al., 1996 [17]	Israel	ADHD N = 43, mean age 10.1 ± 2.4 y.o., female 9.3% (21 with methylphenidate treatment 5–20 mg per day); Controls n = 28, mean age 11.3 ± 3.2 y.o., female 14.3%	Case–control	Exploring serum Zn^2+^ between cases and controls	Lower serum Zn^2+^ in ADHD vs. controls. No association with methylphenidate treatment
Yang et al., 2019 [18]	China	ADHD N = 419, mean age 8.8 ± 2.1 y.o., female 7.9%;Controls n = 395, mean age 8.9 ± 1.7 y.o., female 8.9%	Case–control	Exploring serum Zn^2+^ between ADHD cases and controls; evaluating potential associations of Zn^2+^ levels and ADHD symptoms	Lower serum Zn^2+^ in ADHD vs. controls. Zn^2+^ levels correlated negatively with Swanson, Nolan, and Pelham Rating Scale (SNAP-IV) inattentive subscale (r = −0.40) and total score (r = −0.24).
Tippairote et al., 2017 [19]	Thailand	ADHD n = 45, mean age 5.56 ± 1.34 y.o., female 31%; Controls n = 66, mean age 5.26 ± 1.29 y.o., female 39%	Cross-sectional	Whole blood/serum trace elements incl. Zn^2+^; symptoms	Increased hair Zn^2+^ with more symptoms of inattention, hyperactivity and total ADHD symptoms.
Rosenau et al., 2022 [20]	Netherlands	Children/adolescents with stimulant withdrawal n = 33, mean age 13.9 ± 2.19 y.o., female 24.2% Stimulant continuation group n = 30, mean age 14.1 ± 1.93 y.o., female 20.0%	Observational biomarker analysis of methylphenidate withdrawal on Zn^2+^ levels within a randomised controlled trial	Exploring whether Zn^2+^ may help identify 1) children requiring ongoing methylphenidate treatment 2) exploring Zn^2+^ worth as a viable biomarker 3) the association of Zn^2+^ with ADHD symptoms	Higher baseline Zn^2+^ levels correlated with larger number of errors on the working memory task after withdrawal
Sakhr et al., 2020 [21]	Egypt	Type 1 diabetes + ADHD paediatric cohort n = 60, mean age 10.29 ± 2.99 y.o., female 50%; Controls n = 60, mean age 10.85 ± 2.72 y.o., female 51.7%	Prospective case–control	Exploring the levels of ammonia and various other substances comprising Zn^2+^ in patients with type 1 diabetes mellitus with and without ADHD	Positive correlation between glycosylated haemoglobin and copper/Zn^2+^ ratio in children with type 1 diabetes and ADHD
Salvat et al., 2022 [22]	Spain	ADHD children n = 100 mean age 8.33 ± 2.08 y.o., female 28%; Controls n = 100, mean age 8.26 ± 2.08 y.o., female 28%	Case–control	Explore the pattern of nutrient intake, diets, and anthropometric variables in children with ADHD compared with age-matched controls	Zn^2+^ abnormalities linked with ADHD in T1D
Robinson et al., 2024 [23]	USA	ADHD n = 110, mean age 13.13 ± 0.50 y.o., female 29.1%; Controls n = 173, mean age 13.20 ± 0.60 y.o., female 51.5%	Nested case–control study	Exploring salivary metals in ADHD (comprising Zn^2+^) and in ADHD subtypes—hyperactive, inattentive, combined	Salivary Zn^2+^ levels were associated with higher risk for ADHD combined subtype

Abbreviations: n, sample size; y.o., years old; ADHD, attention deficit hyperactivity disorder; T1D, type 1 diabetes; SNAP-IV, Swanson, Nolan, and Pelham Rating Scale.

**Table 2 ijms-26-10877-t002:** Summary of randomised clinical trials exploring Zn^2+^ supplementation in ADHD.

Author, Year	Country	Sample Features (Mean Age—Range, % Female)	Intervention	Reported Outcomes	Results
Bilici et al., 2004 [24]	Turkey	n = 400, mean age 9.6 ± 1.7 y.o.; 18% female	Zn^2+^ sulphate 150 mg/day, 12 weeks duration—monotherapy	ADHD Scale, Conners Teacher Questionnaire, and DuPaul Parent Ratings of ADHD; serum Zn^2+^ levels increased	Reduced hyperactivity and impulsivity
Akhondzadeh et al., 2004 [25]	Iran	n = 44, mean age 7.9 ± 1.7; 41% female	Methylphenidate + Zn^2+^ sulphate 55 mg/day, 6 weeks, adjunct	Parent and Teacher ADHD Rating Scale	Improved ratings
Arnold et al., 2011 [26]	USA	n = 52, age range 6–14 y.o.	Zn^2+^ glycinate 15–30 mg/day, 8 weeks augmentation to amphetamine	Parent ratings; neuropsychological testing	Allowed for amphetamine dose reduction; equivocal clinical outcomes
Noorazar et al., 2020 [27]	Iran	n = 60, 20% female; 9.6 ± 1.70 y.o.	Zn^2+^ augmentation to methylphenidate, 6 weeks	Conners (total, hyperactivity, impulsivity, inattention)	Improved inattention

Abbreviations: n, sample size; y.o., years old; ADHD, attention deficit hyperactivity disorder.

**Table 3 ijms-26-10877-t003:** Summary of findings of observational studies of Zn^2+^ levels in ASD.

Author, Year	Country	Sample Features (Mean Age—Range, % Female)	Design	Reported Outcomes	Results
Lakshmi Priya et al., 2011 [29]	India	45 with ASD and 50 controls; 4–12 y.o; 20% female among patients	Case–control	Level of trace elements (including Zn^2+^) in hair and nail samples	Significant variation in Zn^2+^ levels among individuals with low-severity Autism as compared with the remaining sample
Li et al., 2014 [30]	China	60 with ASD and 60 controls	Case–control	Level of Serum Zn^2+^ and other elements; Childhood Autism Rating Scale	Lower Zn^2+^ and Zn^2+^/copper ratio in individuals with autism spectrum disorders; lower Zn^2+^/copper ratio associated with higher symptoms severity
Macedoni-Lukšič et al., 2015 [31]	Slovenia	52 children with ASD (average age 6.2 y.o.) and 22 with other neurological disorders (average age 6.6 y.o.)	Case–control	Blood metals; Urine porphyrins	In ASD significantly elevated blood Cu/Zn ratio; no difference in porphyrin levels

Abbreviations: y.o., years old; ASD, autism spectrum disorder.

**Table 4 ijms-26-10877-t004:** Results of the observational study exploring Zn^2+^ levels in Parkinson’s disease.

Author, Year	Country	Sample Features (Mean Age—Range, % Female)	Intervention	Reported Outcomes	Results
Zhao et al., 2013 [38]	China	PD patients n = 238, mean age 66.6 ± 11.3 y.o., 49.2% female; controls n = 302, mean age 65.6 ± 12.2 y.o., 49.3% female	Case–control	Plasma selenium, copper, iron, Zn^2+^	PD patients showed increased plasma Se and Fe, but decreased Cu and Zn compared with controls; lower Zn was associated with increased PD risk

Abbreviations: y.o., years old; PD, Parkinson’s disease.

**Table 5 ijms-26-10877-t005:** Studies investigating the association between alterations of Zn^2+^ homeostasis and AD.

Author, Year	Country	Sample Features (Mean Age–Range, % Female)	Design	Reported Outcomes	Results
Whitfield et al., 2015 [43]	UK, Norway	Postmortem brain samples: AD (n = 15, mean age at death 87 y.o., 67% female), DLB (n = 27), PDD (n = 29), and comparison group without dementia (n = 24)	Clinicopathologic case–control	Brain Zn transporter 3 (ZnT3) expression in Broadmann area 9 with depression severity (neuropsychiatry inventory) in postmortem samples	Reduced Zn^2+^ transporter 3 (ZnT3) in dorsolateral prefrontal cortex associated with higher depression severity across dementia groups, including AD
Rembach et al., 2014 [44]	Australia	AIBL cohort: 1084 participants (AD n = 205, MCI n = 126, controls n = 753); mean age AD 78.8 y.o., controls 70.6 y.o.; 58% female	Cross-sectional cohort	Serum and erythrocyte Zn^2+^	Observed lower serum Zn^2+^ in AD vs. controls, but effect disappeared after age adjustment; Zn^2+^ decline attributed to ageing, not AD
Xu et al., 2018 [45]	UK	AD n = 42, controls n = 43; mean age 78.2 vs. 78.1 y.o.; 52% male	Case–control	Plasma levels of seven metals incl. Zn^2+^; ICP-MS	No overall difference between AD and controls; in males, Zn^2+^ trended higher in AD vs. controls (*p* = 0.021); no difference in females.

Abbreviations: y.o., years old; AD, Alzheimer’s disease; MCI, mild cognitive impairment; AIBL, Australian Imaging, Biomarker, and Lifestyle; ICP-MS, Inductively Coupled Plasma Mass Spectrometry.

**Table 6 ijms-26-10877-t006:** Study selection for papers probing the association of Zn^2+^ metabolism and SCZ.

Author, Year	Country	Sample Features (Mean Age—Range, % Female)	Design	Reported Outcomes	Results
Liu et al., 2015 [46]	China	114 cases, 114 controls—76 pair-males and 38 pair-females. Cases Mean age 32.8 ± 11.3 years and controls 33.0 ± 10.7 years old	Case–control	Serum levels of trace elements in SCZ vs. controls	No evidence for a difference in Zn^2+^ levels between study groups

**Table 7 ijms-26-10877-t007:** Studies exploring the possible association of Zn^2+^ metabolism anomalies in BD.

Author, Year	Country	Sample Features (Mean Age—Range, % Female)	Design	Reported Outcomes	Results
Siwek et al., 2016 [47]	Poland	n = 129 individuals with BD mean age 44.3 ± 12.8 y.o., 46.5% bipolar II; n = 50 controls 72% female, mean age 45.8 ± 12.4 y.o.	Cross-sectional	Zn^2+^ levels in BD patients vs. control and in depending on the mood phase	Lower Zn^2+^ levels in BD type I in bipolar depression vs. other mood phases
Jonsson et al., 2022 [48]	Sweden	n = 121 individuals with BD, female 59.5%, mean age 46.38 ± 1.25 y.o.; n = 30 controls, mean age 46.37 ± 2.80 y.o., 56.6% female	Cross-sectional	Exploring Zn^2+^ blood concentration and its association with Affective Disorder Evaluation, and executive functioning was assessed by using the Delis–Kaplan Executive Function System	Increased Zn^2+^ serum levels unrelated to monocyte chemoattractant protein-1, chitinase 3-like protein 1, and soluble cluster of differentiation 14. No association for Zn^2+^ and executive functioning or symptoms severity
Chebieb et al., 2024 [49]	Algeria	N = 33 individuals with BD, mean age 39.4 ± 11.0 y.o., 51.5% female; n = 38 controls, mean age 40.2 ± 10.9 y.o., female 34%	Case–control, cross-sectional	Exploring Zn^2+^ plasma concentrations and plasma lipid peroxidation (malondialdehyde)	Lower Zn^2+^ in BD patients vs. controls; negative correlation between lipid peroxidation marker and higher copper to Zn^2+^ ratio

Abbreviations: y.o., years old; BD, bipolar disorder.

**Table 8 ijms-26-10877-t008:** Findings of randomised clinical trials describing the potential role of Zn^2+^ in the treatment of MDD.

Author, Year	Country	Sample Features (Mean Age—Range, % Female)	Intervention	Reported Outcomes	Results
Nowak et al., 2003 [50]	Poland	N = 14	Antidepressant + Zn^2+^ augmentation 25 mg/day, 12 weeks vs. placebo	Hamilton Depression Rating Scale, Beck Depression Inventory	Greater symptoms reduction in the active arm
Siwek et al., 2009 [51]	Poland	N = 60, 18–55 y.o., 66% female	Imipramine + Zn^2+^ 25 mg/day augmentation, 12 weeks vs. placebo	Clinical Global Impression, Beck Depression Inventory, Hamilton Depression Rating Scale, Montgomery-Åsberg Depression Rating Scale	No difference between group—possible benefit among treatment resistant patients
Sawada et al., 2010 [52]	Japan	N = 30, female 100%	Multivitamins + Zn^2+^ 7 mg/day vs. Multivitamins	Cornell Medical Index—AL and MR sections for somatic symptoms, mood and feelings	Improved mood subscales; increased Zn^2+^ levels
Ranjbar et al., 2013 [53]	Iran	N = 44, 37 ± 9 in active arm, 37.5 ± 8 in placebo arm	SSRI + Zn^2+^ 25 mg/day augmentation vs. SSRI + placebo	Beck Depression Inventory	Greater mood improvement in the active arm
Ranjbar et al., 2014 [54]	Iran	N = 44	SSRI + Zn^2+^ 25 mg/day augmentation vs. SSRI + placebo	Hamilton depression rating scale; BDNF, cytokines	No changes in biomarkers; greater mood improvements in the active arm
Salari et al., 2015 [55]	Iran	N = 43 MDD in MS	Zn^2+^ sulphate 220 mg/day (~50 mg elemental Zn^2+^) vs. placebo, 12 weeks	Beck Depression Inventory; neurological exam (i.e., abnormal ocular movement, muscle power, and gait disorder)	Greater mood improvements in the active arm, with no effect on the neurological examination
Solati et al., 2015 [56]	Iran	N = 50 with obesity or overweight	Zn^2+^ 30 mg/day vs. placebo, monotherapy	Beck Depression Inventory-II; serum BDNF	Greater mood improvements in the active arm; increased BDNF
Yosaee et al., 2020 [57]	Iran	N = 140 with obesity or overweight	Randomly assigned to one of four groups in a 1:1:1:1 ratio: 2000 IU/d vitamin D + Zn^2+^ placebo; 30 mg/d Zn^2+^ gluconate + vitamin D placebo; 2000 IU/d vitamin D + 30 mg/d Zn^2+^ gluconate; or vitamin D placebo + Zn^2+^ placebo for 12 wk.	Beck Depression Inventory II; serum cortisol; serum BDNF	Zn^2+^ supplements were associated with greater improvements in mood; no cortisol or BDNF effects observed

Abbreviations: y.o., years old; MDD, major depressive disorder; MS, multiple sclerosis; BDNF, brain-derived neurotrophic factor; SSRI, selective serotonin reuptake inhibitor.

**Table 9 ijms-26-10877-t009:** Findings of observational studies testing the role of Zn^2+^ in the treatment of MDD.

Author, Year	Country	Sample Features (Mean Age—Range, % Female)	Design	Reported Outcomes	Results
Maes et al., 1999 [58]	Belgium	n = 48, mean age 54.0 ± 14.1 y.o., female 72%; n = 15 controls, mean age 55.3 ± 13.0 y.o., 53% female	Cross-sectional	Exploring Zn^2+^ serum levels in patients vs. controls	Lower Zn^2+^ and lower albumin in MDD patients vs. controls—at least part of the effect on Zn^2+^ appears mediated by lower albumin
Maserejian et al., 2012 [59]	USA	Boston Area Community Health survey (2002–2005); Centre for Epidemiologic Studies Depression scale defined depressive symptoms; 2163 female and 1545 men	Cross-sectional	Exploring the association of depressive symptoms and self-reported dietary Zn^2+^ intake	Women, but not men, with low dietary Zn^2+^ intake had a higher burden of depressive symptoms
Lehto et al., 2013 [60]	Finland	2317 men, aged 42 to 61 y.o.	Cohort study	Exploring the association of major depression diagnosis at hospital discharge and self-reported dietary Zn^2+^ intake in a 20-year follow-up study, after excluding individuals with high depressive symptoms at baseline	No association with self-reported Zn^2+^ intake was found
Styczeń et al., 2017 [61]	Poland	Patients n = 114; mean age 49.4 ± 10.7 y.o.; female 75%; Controls n = 50; mean age 45.8 ± 12.4 y.o.; 72% female;	Cross-sectional study	Exploring plasma Zn^2+^ levels association with treatment outcomes in major depressive disorder	Serum Zn^2+^ levels were lower among individuals with MDD as compared with controls; individuals reaching symptomatic remission had Zn^2+^ levels similar to controls
Islam et al., 2018 [62]	Bangladesh	Patients n = 247; mean age 33 ± 0.6 y.o.; 63% female. Controls n = 248; mean age 33 ± 0.6 y.o.; 59% female	Case- control study	Exploring the levels of macrominerals (calcium and magnesium) and trace elements (copper, manganese, selenium, iron and Zn^2+^) in MDD vs. controls	Significantly increased copper and decreased levels of magnesium, calcium, manganese, selenium and Zn^2+^ in MDD compared to controls

Abbreviations: MDD, Major Depressive Disorder; y.o., years old.

**Table 10 ijms-26-10877-t010:** Summary of recommended Zn^2+^ daily doses and possible risks associated with toxicity [65].

Patient Group	RDA/AI (mg/Day)	UL (mg/Day)	Higher Intake Thresholds	Main Adverse Effects
*Infants*				
0–6 months	2	4	—	—
7–12 months	3	5	—	—
*Children*				
1–3 years	3	7	—	—
4–8 years	5	12	—	—
Adolescents (9–18 y)	Males: 8–11; Females: 8–9	23–34	—	—
Adults (≥19 y)	Males: 11; Females: 8; Pregnancy: 11–12; Lactation: 12–13	40	Up to ~100 mg/day generally tolerated	GI upset (nausea, vomiting, diarrhoea, abdominal pain)
General adults—excess	—	—	>150 mg/day	Copper deficiency due to impaired absorption
Extreme excess (mega-doses/contamination)	—	—	Very high intakes	Severe GI symptoms: in rare cases calcium disodium ethylenediaminetetraacetate (CaNa2EDTA) chelation is used

Abbreviations: RDA, Recommended Dietary Allowance; AI, Adequate Intake; UL, Upper intake level; GI, gastrointestinal. UL = 40 mg/day (≥19 y) for adults refers to elemental Zn^2+^; monitor copper with long-term doses ≥ 25–30 mg/day.

## Data Availability

No new data were created or analyzed in this study. Data sharing is not applicable to this article.

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
