# Peer review of "Perturbations of Zinc Homeostasis and Onset of Neuropsychiatric Disorders"

_ijms, 2025, doi:10.3390/ijms262210877_

Round 1
Reviewer 1 Report
Comments and Suggestions for Authors
The manuscript ‘Perturbations of zinc homeostasis and onset of neuropsychiatric disorders’ explores the impact of zinc on neuropsychiatric disorders. It does so in a clear manner, using correct language and useful tools such as tables. Thanks to the well-written introduction, we learn about the role of Zn in the organism. The chapter on methodology clearly defines the method of selecting articles for this review. However, despite its many strengths, the manuscript has several weaknesses that, if eliminated, would improve its value:
- The text needs to be standardised in terms of the use of element names and abbreviations. I suggest using the full name of the element with the abbreviation in () the first time it appears, and then only the abbreviation thereafter.
- There is a missing list of abbreviations with explanations at the end of the manuscript.
- The structure of the text needs improvement: I do not understand why the levels and role of zinc in ADHD are discussed in Chapter 3 and not in a separate chapter, like other diseases. As a solution to this problem, I suggest creating Chapter 3. The role and levels of Zn in neurodevelopmental disorders, and subsections 3.1 The role and level of Zn in ADHD, 3.2 The role and level of Zn in ASD, etc.
- I wonder why the font size of the paragraph about ASD is smaller? I also suggest moving the sentence from lines 97-100 to this paragraph: "Indeed, a study on trace element concentration carried out in children with ASD, measuring the content of essential trace elements in hair samples, found that the most deficient element was zinc [13]. Zinc deficiency was found in 92% of affected children as compared with 20% of healthy subjects [13]."
- Abbreviations are not always used correctly: line 181 - The full name of the disease is not required; PD should be used.
- Line 192 – I don't understand why [33–36] is in bold. ANK3 gene in line 301 should be written in italics.
- The numbering of tables needs to be corrected: Tables 4 and 11 are missing. They should also be referenced appropriately in the text of the article.
- Table 12 should be corrected – it appears to have been prepared inconsistently.
- Citing articles in full in the text needs improvement – after using the phrase ‘et al.’, the article number from the bibliography should also appear at the end of the sentence referring to that article, and not just after the phrase ‘et al.’.
After correcting the above-mentioned issues, the manuscript should be suitable for publication.

Author Response
The manuscript “Perturbations of zinc homeostasis and onset of neuropsychiatric disorders’ explores the impact of zinc on neuropsychiatric disorders. It does so in a clear manner, using correct language and useful tools such as tables. Thanks to the well-written introduction, we learn about the role of Zn in the organism. The chapter on methodology clearly defines the method of selecting articles for this review.
- R) We thank the reviewer for the positive assessment of our manuscript.
Q1) However, despite its many strengths, the manuscript has several weaknesses that, if eliminated, would improve its value.
R1) We have extensively revised the text to encounter the requests of the reviewer.
Q2) The text needs to be standardised in terms of the use of element names and abbreviations. I suggest using the full name of the element with the abbreviation in () the first time it appears, and then only the abbreviation thereafter.
R2) This was correct throughout the text.
Q3) There is a missing list of abbreviations with explanations at the end of the manuscript.
R3) This was added at the end of manuscript.
Q4) The structure of the text needs improvement: I do not understand why the levels and role of zinc in ADHD are discussed in Chapter 3 and not in a separate chapter, like other diseases. As a solution to this problem, I suggest creating Chapter 3. The role and levels of Zn in neurodevelopmental disorders, and subsections 3.1 The role and level of Zn in ADHD, 3.2 The role and level of Zn in ASD, etc.
R4) This was done in the revised version of the manuscript.
Q5) I wonder why the font size of the paragraph about ASD is smaller? I also suggest moving the sentence from lines 97-100 to this paragraph: "Indeed, a study on trace element concentration carried out in children with ASD, measuring the content of essential trace elements in hair samples, found that the most deficient element was zinc [13]. Zinc deficiency was found in 92% of affected children as compared with 20% of healthy subjects [13]."
R5) This has been corrected according to the reviewer suggestions.
Q6) Abbreviations are not always used correctly: line 181 - The full name of the disease is not required; PD should be used.
R6) This has been corrected.
Q7) Line 192 – I don't understand why [33–36] is in bold. ANK3 gene in line 301 should be written in italics.
R7) This has been corrected.
Q8) The numbering of tables needs to be corrected: Tables 4 and 11 are missing. They should also be referenced appropriately in the text of the article.
R8) This has been revised and the Tables are now numbered correctly.
Q9) Table 12 should be corrected – it appears to have been prepared inconsistently.
R9) This was revised accordingly.
Q10) Citing articles in full in the text needs improvement – after using the phrase ‘et al.’, the article number from the bibliography should also appear at the end of the sentence referring to that article, and not just after the phrase ‘et al.’.
R10) Revised
Reviewer 2 Report
Comments and Suggestions for Authors
The review of Faa et al provides a comprehensive overview of the role of zinc in various central nervous system disorders, as well as the potential implications of its supplementation or, more broadly, the restoration of proper metal homeostasis. The data presented underscore the need for further in vivo studies to substantiate the in vitro findings. Overall, the work is of interest and deserves publication after minor revisions (see comments below).
- One limitation of covering such a wide range of pathologies is the risk of underestimating the literature on some of them. In particular, the paragraph on Alzheimer’s disease does not consider several studies conducted in animal models that demonstrate the role of zinc in the recovery of cognitive abilities or in slowing the progression of dementia (e.g., DOI: 1038/s41598-025-97830-6 ). A more extensive discussion on this pathology is necessary to provide a broader and more balanced view.
- The manuscript should be improved in terms of English language usage (e.g. line 97-99 verb is missing “ a study,……, showed?...; line 303-305 rephrase”.
Author Response
The review of Faa et al provides a comprehensive overview of the role of zinc in various central nervous system disorders, as well as the potential implications of its supplementation or, more broadly, the restoration of proper metal homeostasis. The data presented underscore the need for further in vivo studies to substantiate the in vitro findings. Overall, the work is of interest and deserves publication after minor revisions (see comments below).
- R) We thank the reviewer for the positive assessment of our manuscript.
Q1) One limitation of covering such a wide range of pathologies is the risk of underestimating the literature on some of them. In particular, the paragraph on Alzheimer’s disease does not consider several studies conducted in animal models that demonstrate the role of zinc in the recovery of cognitive abilities or in slowing the progression of dementia (e.g., DOI: 1038/s41598-025-97830-6). A more extensive discussion on this pathology is necessary to provide a broader and more balanced view.
R1) This reference was added to the manuscript in the Discussion. We would like to highlight that the aim of our review was to identify perturbations of zinc as modulators of risk of neuropsychiatric disorders in humans; thus animal studies were not included in the results. In addition, given the narrative nature of the review, we are aware (and have acknowledged) that some evidence might have not been identified with our search strategy. Thus, we added the following sentences (Discussion) including also the limitations: “Interestingly, novel approaches could include the production of novel zinc-based drug substance that may show more effective properties in modulating the biological per-turbations associate with neuropsychiatric disorders. Indeed, a recent study used KLS-1, which is zinc aspartate enriched with light isotope 64Zn to 99.2% atomic fraction of total zinc, in an animal model of AD, showing a decreased inflammatory load in the CNS of rats that correlated with improvement of short-term spatial memory and cognitive flexibility, and moderately—with betterment of remote spatial memory [61]. Finally, we should highlight that this review is limited by the application of a non-systematic approach. Thus, evidence pertinent to the role of Zn2+ in neuropsychiatric disorders might not have been identified exhaustively making our indications preliminary and in need of integration with other sources of data.”
Q2) The manuscript should be improved in terms of English language usage (e.g. line 97-99 verb is missing “ a study,……, showed?...; line 303-305 rephrase”.
R2) We rephrased these sentences as requested.